# High Proportion of Potential Candidates for Immunotherapy in a Chilean Cohort of Gastric Cancer Patients: Results of the FORCE1 Study

**DOI:** 10.3390/cancers11091275

**Published:** 2019-08-30

**Authors:** Miguel Cordova-Delgado, Mauricio P. Pinto, Ignacio N. Retamal, Matías Muñoz-Medel, María Loreto Bravo, María F. Fernández, Betzabé Cisternas, Sebastián Mondaca, César Sanchez, Hector Galindo, Bruno Nervi, Carolina Ibáñez, Francisco Acevedo, Jorge Madrid, José Peña, Erica Koch, Maria José Maturana, Diego Romero, Nathaly de la Jara, Javiera Torres, Manuel Espinoza, Carlos Balmaceda, Yuwei Liao, Zhiguang Li, Matías Freire, Valentina Gárate-Calderón, Javier Cáceres, Gonzalo Sepúlveda-Hermosilla, Rodrigo Lizana, Liliana Ramos, Rocío Artigas, Enrique Norero, Fernando Crovari, Ricardo Armisén, Alejandro H. Corvalán, Gareth I. Owen, Marcelo Garrido

**Affiliations:** 1Hematology & Oncology Department, Faculty of Medicine, Pontificia Universidad Católica de Chile (PUC), Santiago 8330032, Chile; 2Faculty of Chemical & Pharmaceutical Sciences, Universidad de Chile, Santiago 8380494, Chile; 3Department of Physiology, Faculty of Biological Sciences, PUC, Santiago 8331150, Chile; 4Faculty of Dentistry, Universidad de Los Andes, Santiago 7591538, Chile; 5Department of Pathology, Faculty of Medicine, PUC, Santiago 8330023, Chile; 6Advanced Center for Chronic Diseases (ACCDiS), Core Biodata, Santiago 8330034, Chile; 7Centre of Clinical Research, Health Technology Assessment Unit, PUC, Santiago 8330032, Chile; 8Department of Public Health, PUC, Santiago 8330032, Chile; 9Center of Genome and Personalized Medicine, Institute of Cancer Stem Cell, Dalian Medical University, Dalian 116044, Liaoning, China; 10Center of Excellence in Precision Medicine (CEMP), Obispo Espinoza Campos 2526, Macul, Santiago 7810305, Chile; 11Department of Gastrointestinal Surgery, PUC, Santiago 8330024, Chile; 12Biomedical Research Consortium of Chile, Santiago 8331010, Chile; 13Millennium Institute on Immunology and Immunotherapy, Santiago 8331150, Chile

**Keywords:** gastric cancer, gastric adenocarcinoma, cancer subtypes, prognosis, survival, molecular

## Abstract

Gastric cancer (GC) is a heterogeneous disease. This heterogeneity applies not only to morphological and phenotypic features but also to geographical variations in incidence and mortality rates. As Chile has one of the highest mortality rates within South America, we sought to define a molecular profile of Chilean GCs (ClinicalTrials.gov identifier: NCT03158571/(FORCE1)). Solid tumor samples and clinical data were obtained from 224 patients, with subsets analyzed by tissue microarray (TMA; *n* = 90) and next generation sequencing (NGS; *n* = 101). Most demographic and clinical data were in line with previous reports. TMA data indicated that 60% of patients displayed potentially actionable alterations. Furthermore, 20.5% were categorized as having a high tumor mutational burden, and 13% possessed micro-satellite instability (MSI). Results also confirmed previous studies reporting high Epstein-Barr virus (EBV) positivity (13%) in Chilean-derived GC samples suggesting a high proportion of patients could benefit from immunotherapy. As expected, *TP53* and *PIK3CA* were the most frequently altered genes. However, NGS demonstrated the presence of *TP53*, *NRAS*, and *BRAF* variants previously unreported in current GC databases. Finally, using the Kendall method, we report a significant correlation between EBV+ status and programmed death ligand-1 (PDL1)+ and an inverse correlation between p53 mutational status and MSI. Our results suggest that in this Chilean cohort, a high proportion of patients are potential candidates for immunotherapy treatment. To the best of our knowledge, this study is the first in South America to assess the prevalence of actionable targets and to examine a molecular profile of GC patients.

## 1. Introduction

Worldwide, stomach or gastric cancer (GC) currently ranks as the fifth most common malignancy and the third leading cause of cancer mortality [1]. Studies demonstrate GC incidence and mortality rates display wide regional/geographical heterogeneity. Over half of the new GC cases are diagnosed in developing countries. High-risk areas include East Asian, Eastern Europe, Central and South American countries. In contrast, Southern Asia, North and East Africa, North America, Australia, and New Zealand are low-risk areas. Similarly, mortality/survival rates exhibit a wide geographical heterogeneity. Within South America, mortality rates range from 3.6 to 23.6 per 100,000/year. In Chile, GC is the leading cause of cancer death with 25.1 and 13.2 per 100,000/year for men and women, respectively [2,3]. Although several factors may explain this scenario, most studies suggest that high mortality rate can be attributed to late diagnosis, due to slow-growing asymptomatic tumors.

Histopathologically, GCs can be classified as intestinal- or diffuse-type, according to Lauren. In general, diffuse tumors are characterized by worse prognosis compared to intestinal-type; they are also characterized by a lower benefit derived from adjuvant radiotherapy. Beyond this, the influence of histological-type upon patient treatment decisions remains undetermined. More recently, the World Health Organization (WHO) proposed an alternative system [4]. Even with this new system, several cases that display similar histology have disparate treatment responses and/or prognoses. Indeed, current GC classification systems provide little, if any relevant information for clinical management of patients. Therefore, a clinically meaningful classification that aids the selection of more effective treatments for each patient is urgently needed.

To date, at least three major studies have sought to define GC molecular subtypes using patient cohorts [5,6,7]. Although these studies have successfully defined GC molecular subtypes based on expression profiles, hot-spot mutations, genomic rearrangements, and micro-satellite instability (MSI), the association of molecular subtypes to clinical parameters and/or patient outcomes remains unclear. 

Another characteristic feature of GC is its association with infectious agents, such as *Helicobacter pylori* or the Epstein-Barr virus (EBV). As occurs with GC incidence, the distribution of histological types and the frequency of both *H. pylori* and EBV are highly heterogeneous across the globe. In Chile, >70% of the population is *H. pylori*+ [8], while approximately a 16% of GCs are EBV+ [9,10], a percentage much higher than the observed in most countries [9,11,12].

Here, we report the results of the Chilean Gastric Cancer Task Force One (FORCE1), a collaborative pioneer initiative in South America which aimed to profile a cohort of 224 Chilean GC patients (Clinicaltrials.gov identifier: NCT03158571) [13]. Our study included clinical, demographics, protein expression, and molecular profiling data. We report patient survival by gender, stage, and Lauren histological type. Finally, we compare our results to GC databases from The Cancer Genome Atlas (TCGA). To the best of our knowledge, this is the first prevalence study on “actionable” genes in South American patients. Our study seeks to provide a first indication on the percentage of patients who could benefit from targeted therapies, helping to delineate future prevention strategies against GC.

## 2. Results

### 2.1. Patients’ Basic and Demographic Information

Demographic and clinical-pathological characteristics of patients are summarized in Table 1. Patients were predominantly males (63.4%) and classified at stages III/IV (61.2%). Most tumors were located at the stomach corpus (38.4%). Histologically, 33.9% of samples were intestinal-type, and 76% were adenocarcinomas.

### 2.2. Expression Profiling of Tumor Samples

We performed expression profiling by tissue microarray (TMA) in a subset of samples. Using immunohistochemistry (IHC) we evaluated PDL1 the status of four mismatch repair (MMR) proteins, namely MLH1, PMS2, MSH2, and MSH6. In addition, HER2, p16, and p53 were determined by IHC. EBV status was determined by chromogenic in situ hybridization (CISH, Table 2). We found that 28.9% of patients in our cohort were PDL1+ (by combined positive score (CPS) ≥ 10); 13% were classified as MMR-deficient (suggesting MSI-H). Our data showed that 13.3% of patients were HER2+. Previous studies demonstrate that p16 and p53 are tumor suppressor genes frequently inactivated/mutated in GC; here we found the absence of p16 expression in 36.7% and 53.3% of patients displayed p53 expression (suggesting mutation). Finally, our analysis indicated 13.3% of cases were EBV+. Appendix A summarizes clinical and pathological characteristics of patient subsets: HER2+, EBV+, PDL1+, and MSI+. Notably, EBV+ patients were younger versus other subsets, predominantly male (11 out of 12) and displayed an absence of signet-ring cells. On the other hand, 8 out of 13 MSI+ patients were proximally-third gastric cancers, and six of them were classified as diffuse by Lauren, five of them were of signet-ring cells. Finally, 8 out of 12 HER2+ patients were classified as intestinal by Lauren, and five were signet-ring cell+.

### 2.3. Patient Survival Rates

Patient overall survival (OS) rates were calculated as at 1 June 2018. Survival curves are shown in Figure 1. From the initial group of 224, clinical data were available for 220 patients; median OS for the entire group was 39 months (Figure 1A). Males displayed better OS rates versus females. However, these differences did not reach statistical significance (Log Rank *p* = 0.54, Figure 1B). As expected, cancer stage had a significant impact on OS: 30 or 13 months for stage III or IV, respectively, and 62 months for stage II. Stage I patients did not reach 50% survival (Figure 1C, Log Rank *p* = 0.0001). Lastly, we evaluated OS rates by histological type. We found that the median OS for the intestinal-type was higher than diffuse-type: 42 versus 26 months, respectively. Again, these differences did not reach statistical significance (Log Rank *p* = 0.42. Figure 1D).

### 2.4. Next Generation Sequencing

A total of 143 cancer-related genes were analyzed using the Oncomine comprehensive assay v1 [13] in 101 patients who passed quality controls (QCs); within this subset, 48 had complete datasets and are shown in Figure 2. The waterfall plot shows that the *TP53* gene was the most frequently altered. The upper section shows the number of alterations per patient. Among single nucleotide variants (SNVs), the most frequent alteration was missense *TP53* mutation: 49%; among copy number variants (CNVs), *MYC* amplification was present in 4.9% (Figure 2 and Table 3).

### 2.5. Mutation Prevalence, Comparison with the TCGA Database

Next, we analyzed the most prevalent aminoacidic changes in the 10 most frequently altered genes (*TP53, PIK3CA, VHL, NRAS, KRAS, BRAF, RHOA*, and *APC*). Table 4 shows a total of eight mutations in *TP53* (seven missense and one nonsense) and five missense in *PIK3CA.* All other genes had one mutation associated, except *NRAS* (Table 4). Then, we compared our results with the reported prevalence in four different cohorts from the TGCA database [6,14,15,16]. The most frequent *TP53* mutation in our study was R273C/H (3.9%). The lollipop plot in Appendix A shows the location of *TP53* mutations in our study.

Appendix A summarizes driver mutations found in our cohort; overall, a total of 22 driver mutations were found in 35 patients. Furthermore, we found 32 potentially actionable alterations. Appendix A summarizes affected genes, actionable mutations, associated drug (s), levels of evidence cancer type, and the number of affected patients within our cohort.

### 2.6. Correlation Matrix

Finally, we performed an exploratory analysis. We elaborated two correlation matrices: clinical data/TMA (Figure 3a) and clinical data/NGS correlations (Figure 3b). Given the exploratory nature of these analyses, we did not adjust by multiple comparisons. Correlational analyses used the Kendall method, and only significant correlations (*p* < 0.05) are highlighted.

## 3. Discussion

As with other malignancies, GC is a highly heterogeneous disease. This applies not only to GC histopathological and phenotypic profiles but also to its geographical variations in incidence and mortality. Hence, a better patient stratification based on molecular profiling obtained from different geographical areas might contribute to rationalize the use of targeted therapies, improving patient survival. Our study found that most basic and demographic patient data in our cohort were comparable to previous reports indicating high prevalence among males [17], and diagnosis at an advanced stage (i.e., stages III/IV) [18]. As described previously, we found a high rate of p16 loss [19]. In line with previous reports, we also found 13% of patients were MSI+ (inferred from MMR deficiency) [20], however, within the MSI+ subgroup, we noticed several cases that were classified as diffuse histology by Lauren (6/13: 46%). In contrast, the TCGA study found that most diffuse cancers are classified as “Genomically-Stable” [6], and the percentage of diffuse cancers in the MSI-H subgroup is rather low (about 10%). We speculate this discrepancy can be attributed to technical differences between our study and the TCGA and to the inherent inaccuracy of classification systems based on histological features, such as Lauren. On the other hand, 13% of patients in our cohort were EBV+, suggesting this subset could include candidates for immunotherapy. Median OS for the entire group was 39 months, and the lowest median OS was observed in stage IV patients (Figure 1). 

To date, TNM staging remains the gold standard for GC prognosis and patient survival. Over recent decades, several studies have sought to define a molecular classification for GC [5,6,7,14,21]; however, the clinical utility and applicability of these systems remains limited. Herein, we present NGS data for a subset of 101 patients. A recent study by Ichikawa et al. [22] elaborated a comprehensive genomic profile of GC patients using an NGS panel of 435 genes, including 69 actionable genes; they also determined EBV and MSI status. A total of 207 Japanese patients were divided into seven hierarchical clusters. First, they divided patients in hypermutated (*n* = 32; 15.45%) and non-hypermutated (*n* = 175; 84.54%). Then, non-hypermutated tumors were subdivided into six categories by alterations in: ERBB2 (*n* = 25; 12.07%), CDKN2A/B (*n* = 10; 4.83%), KRAS (*n* = 10; 4.83%), BRCA2 (*n* = 9; 4.34%), ATM (*n* = 12; 5.79%), and a cluster 6 (*n* = 109; 52.6%) with minor or no alterations in actionable genes. Our NGS found 150 SNVs, 31 CNVs, and 9 fusion drivers. We also found 22 driver mutations, and 32 patients had mutations on 14 actionable targets out of 101 analyzed (31.68%, see Appendix A). In line with the abovementioned study, we found *ERBB2* (*n* = 5; 5.0%), *CDKN2A* (*n* = 4; 4.0%), *KRAS* (*n* = 8; 7.9%), and *ATM* (*n* = 2; 2.0%) were among the most frequently altered (including both SNVs and CNVs) genes in our cohort (shown in Table 3). Previously, studies have postulated high-tumor mutational burden (TMB) as a predictor biomarker for immunotherapy response in several cancers [23], including advanced GC [24]. However, a recent study found that a high-TMB was not associated with response to nivolumab in GC patients [25]. Previous reports have also shown that estimated TMB values based on targeted gene panel sequencing can display a high degree of discordance versus TMB calculated from whole-exome sequencing [26]. Here, we estimated TMB (see methods and Appendix A) and found a median value = 3.84 mut/Mb. Hence, using an 8.8 mut/Mb cutoff value [27], we found a 21/101 (20.79%) of our patients could be categorized as “highly mutated” and could be candidates for immunotherapy. Interestingly, we observed that 4/10 (40%) of MSI+ patients could be classified as high-TMB, in contrast only 7/59 (11.9%) of microsatellite stable (MSS) patients were high-TMB using the aforementioned cutoff value (Appendix A). In line with our findings, a recent report in Chinese GC patients estimated TMB from a 381-gene panel and found a median TMB = 4.03 in gastric carcinomas [28].

As expected, our NGS data confirmed the *TP53* gene as the most frequently altered among Chilean patients. This is in line with previous reports for various malignancies [29], including GC [22,30]. Accordingly, our TMA data indicate 53.3% of p53 mutants (Table 2), a percentage further confirmed by NGS, indicating 49% of patients displayed *TP53* mutations. As explained above, an effective GC patient stratification should identify biomarkers to select potential responders to targeted therapies. Here we elaborated correlation matrices (Figure 3a,b) searching for potential associations. Overall, >50 statistically significant correlations among the variables measured in our study were discovered using the Kendall method; some of these could serve for patient stratification. Firstly, we found a positive correlation between EBV+ and PDL1 expression (Figure 3a), an association previously described by others [31]. Although EBV+ GCs are generally associated with better prognosis [32,33], a recent article suggests a poorer prognosis in EBV+ GC by high intra-tumoral PDL1 expression [34]. While it is still controversial, PDL1 expression has been postulated as a biomarker for immunotherapy responsiveness [35], and therefore, our data would support the use of PD1-PDL1 checkpoint inhibitors in EBV+ GCs as a first or second-line therapy; a notion confirmed by previous studies that recommend the use of immunotherapeutic drugs against EBV+ subtype GC tumors defined by the TCGA [36]. Second, the presence of signet ring cells (SRCP) is a poor prognostic marker in GC [37]. Accordingly, we found SRCP was negatively correlated to EBV+ and PDL1 status in our series. Regarding p53, we found negative associations with MSI+ (Figure 3a) and also between TP53 and PIK3CA mutations (Figure 3b), In fact, studies in colorectal cancers indicate p53 mutation is commonly associated to MSS/*BRAF* mutant tumors [38], again this partially confirms our findings. On the other hand, *BRAF* mutations in our cohort correlated with alterations in multiple genes including the EGFR–EGFR fusion, correlated SNVs were: *KRAS*, *FBXW7*, *WT1*, *EGFR*, *KDR*, and *ERBB3* (see Figure 3b) suggesting BRAF could be an actionable target for a subset of GC patients who also carry somatic EGFR and/or KRAS alterations. Several studies demonstrate V600E is the most frequently observed BRAF mutation, particularly in melanomas [39]. However, most of our patients displayed the D594G variant, a missense gain-of-function mutation located in the BRAF kinase domain. Interestingly, this *BRAF* variant remains largely unreported by other TCGA GC cohorts (Table 4). Studies suggest D594G is insensitive to conventional BRAF inhibitors that target the V600E mutation, such as Vemurafenib [40]. Recently, several studies have reported a novel generation of BRAF inhibitors that may serve for future therapeutic interventions in GC patients who harbor the D594G mutation [41,42]. Our analysis also found an association between β-catenin (*CTNNB1*) mutations and alterations in *GATA2*, *HRAS*, and HER2 (*ERBB2*). Previous studies have implicated Wnt/β-catenin signaling in gastric tumorigenesis, progression, and metastasis [43]. Interestingly, constitutive Wnt/β-catenin signaling usually results from *H. pylori* infection; in Chile, it is estimated that >70% of the population is *H. pylori+* [8]. Therefore, it is reasonable to speculate this could be an environmental factor that plays a role in the country’s high incidence of GC. Further, a study by Khalil et al. used a tissue microarray and postulated a role of the Wnt/β-catenin pathway in the early stages of HER2+ breast neoplasias [44], suggesting this pathway could also serve for future targeted therapies in GC, especially for HER2+ GC cases.

Given its observational nature, our study has certain limitations: first, although the total number of patients with clinical data is in line with similar reports (*n* = 224) IHC and NGS analyses were performed in patient subsets (*n* = 90 and *n* = 101, respectively) that may represent a bias of our study. Similarly, our cohort reports a relatively high number of stage III cases; unlike stage IV, these patients usually go through surgery providing tissue samples for NGS/TMA analyses, some of our research team members are gastric surgeons, and consequently, this may be a registration bias of our study. As explained, tumor mutational burden (TMB) was estimated from our targeted-gene panel, these estimates may differ significantly from TMB values calculated from whole-exome sequencing and therefore could limit the scope of our findings. 

## 4. Materials and Methods

### 4.1. Patients, Ethics Approval, Consent to Publish and Demographic Data

A total of 224 GC patients diagnosed between April 2004 and March 2018 were enrolled at the Centro de Cancer UC-CHRISTUS in the Pontificia Universidad Catolica de Chile (PUC). Inclusion criteria: age ≥18, diagnosed with confirmed GC, with clinical follow-up, able to understand spoken and written Spanish and sign an informed consent. The Internal Review Board and the Ethics and Scientific Committee at the School of Medicine approved this research (#16-046 dated 21 April 2016). Participants signed a consent to publish forms. A waiver of consent was granted to include deceased patients into the study. All patient data were anonymized and coded.

### 4.2. Protein Expression Analyses by Tissue Microarray (TMA) and EBV Status

PDL1, MLH1, PMS2, MSH2, MSH6, HER2, p16, and p53 expression was determined in 90 patients using manually prepared TMA [45,46] from deparaffinized sections obtained from archival samples of cancer tissues. Antibodies were PD-L1: Cat # SK00521-k (Dako, Carpinteria, CA, USA); all other antibodies: MLH1: Cat # 06472966001; PMS2: Cat # 06419216001; MSH2: Cat # 05269270001; MSH6: Cat # 5929911001); HER2: Cat # 05278368001; p16: Cat # 06695221001; p53: Cat # 5278074001 were from Roche Diagnostics (Basel, Switzerland). PDL1+ status was determined from the combined positive score (CPS); this score assesses the proportion of PDL1+ tumor cells and PDL1+ tumor-associated cells divided by the total number of cells ×100 [47]. In our study, we set a cutoff CPS ≥10 for PDL1+ patients following the recommendation of the upcoming KEYNOTE062 study (ClinicalTrials.gov identifier: NCT02494583). HER2+ status was assessed following a protocol described previously [48]. Briefly, HER2+++ by IHC analysis were considered positive, HER2++ cases were further confirmed by fluorescence in situ hybridization (FISH). Samples that were + or 0 by IHC were considered negative. In addition, EBV status was determined by EBER1 expression in infected cells by chromogenic in situ hybridization (CISH) method as described [9] with slight modifications.

### 4.3. Next Generation Sequencing (NGS), DNA/RNA Purification & Quantification 

The Oncomine comprehensive assay v.1 kit (Thermo Fisher, Carlsbad, CA, USA) was used [13] in a subset of 116 patients. Nucleic acids (DNA/RNA) were extracted from formalin-fixed paraffin-embedded (FFPE) samples using the RecoverAll kit (Cat #AM1975, Thermo Fisher). Samples were quantified in a Qubit Fluorometer 3.0™ (Thermo Fisher). For DNA and RNA quantifications Qubit™ dsDNA/RNA HS Assay (Thermo Fisher) was used.

### 4.4. Construction, Quantification, and Sequencing of Libraries

Libraries were constructed according to the “Ion Ampliseq™ Library Preparation protocol using the Oncomine Cancer Research Panels” and were quantified by qPCR. The pool of libraries required for enrichment in the Ion Chef™ System (Thermo Fisher) which was performed in the ratio 8:2 (DNA:RNA) for each sample to be analyzed; each pool of libraries consists of up to 8 samples for analysis/chip. The enrichment of the libraries and subsequent loading of the chip was done in the Ion Chef™ System. Libraries were sequenced on the Ion S5™ NGS system (Thermo Fisher).

### 4.5. Bioinformatics Methods

Genomic analyses were performed in 101 and 86 samples that passed quality controls (QCs; described below) for DNA and RNA, respectively. Genomic data were processed in two stages; first, variant calling was performed using Thermo Fisher platforms: Torrent Browser and Ion Reporter with the Oncomine Focus w2.1 workflow. Subsequent analyses were performed using in-house developed methods. Then short (<25 bp) and low quality (Phred Scale, Q < 16) reads were eliminated; and the remaining reads were demultiplexed. Subsequently, DNA samples were aligned against the reference human genome (Hg19 version) defined by the hotspot browser extensible data (BED) file. High-quality reads followed the analysis defined by Oncomine, where the variant calling in both DNA and RNA was performed. Annotation of detected alterations was performed by Oncomine Variant Annotator plugin 2.2.7 (OVA), supported by dbSNP, ClinVar, and VariantDB databases. QC was performed following the manufacturer´s suggestions. All information was summarized in Appendix A. Tumor mutational burden (TMB) was estimated from the number of somatic non-synonymous variants per Megabase (mt/Mb) identified at the hotspot regions interrogated by the Oncomine Comprehensive v1 panel (Appendix A).

### 4.6. Identification of Drivers and Druggable/Actionable Mutation

Our data were compared to the comprehensive report by Bailey et al. [49], and to establish the number of druggable/actionable mutations, we downloaded the oncoKB [50] list of actionable variants (Last update: January 24th, 2019). This list contains a series of somatic alterations associated with specific therapies, across different levels of evidence and cancer types. Evidence level ranges from 1 (highest) to 4 (lowest). R1 and R2 levels are related to predictive resistance to treatment.

### 4.7. Statistical Analysis

Survival analysis was performed using the Kaplan–Meier method and the log–rank test. Correlation analyses were performed using the Kendall method (for non-parametric, categorical, and continues variables). *p* < 0.05 *p*-values were considered statistically significant. All data were analyzed using R software (The R Foundation, Vienna, Austria).

## 5. Conclusions

Our study analyzed clinical, genomic, transcriptomic, and protein expression data in a cohort of Chilean GC patients. Most basic and demographic data were in line with previous GC reports. However, our NGS data indicated the presence of novel *TP53*, *NRAS*, and *BRAF* variants not reported on GC databases. To the best of our knowledge, this is the first prevalence assessment of actionable targets among Chilean GC patients. Finally, our data suggest that a high proportion of patients, including EBV+ (13%), MSI+ (13%) and high-TMB (20%) patients, could benefit from the use of immunotherapeutic drugs.

## Figures and Tables

**Figure 1 cancers-11-01275-f001:**
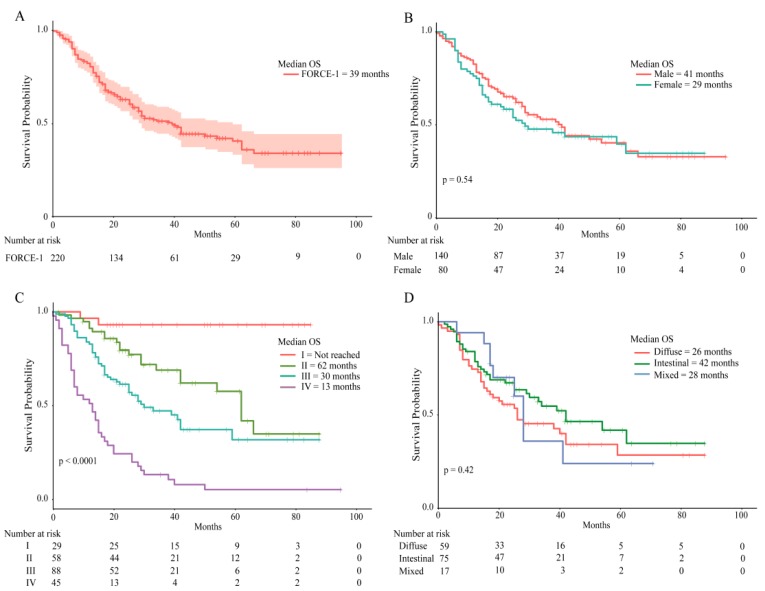
Overall survival rates in the FORCE1 cohort. Kaplan–Meier curves indicate overall survival for (**A**) the entire cohort, (**B**) by gender, (**C**) by cancer stage, (**D**) by histological type.

**Figure 2 cancers-11-01275-f002:**
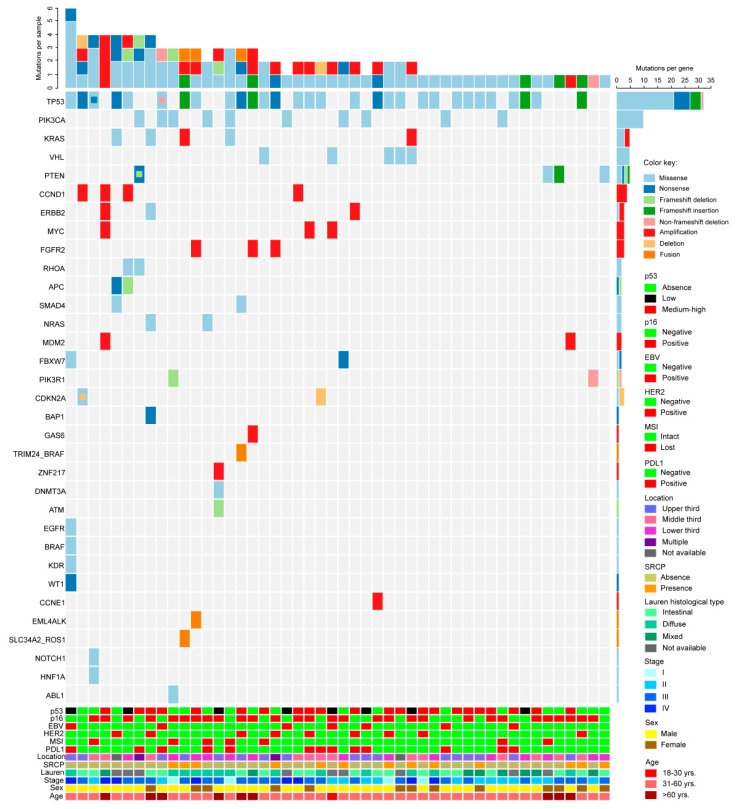
Profiling of 48 Chilean gastric cancers by next generation sequencing (NGS), clinical, and pathological characteristics. The waterfall plot shows the number of gene alterations per patient (upper section), number of alterations per gene (right). Colored squares indicate the alteration type (SNV, CNV, or fusion drivers/see key). Clinical (age, gender, Lauren classification, signet ring, tumor location), and pathological (PDL1, MSI, HER2, p16, p53, EBV) characteristics for each patient are shown in the lower section.

**Figure 3 cancers-11-01275-f003:**
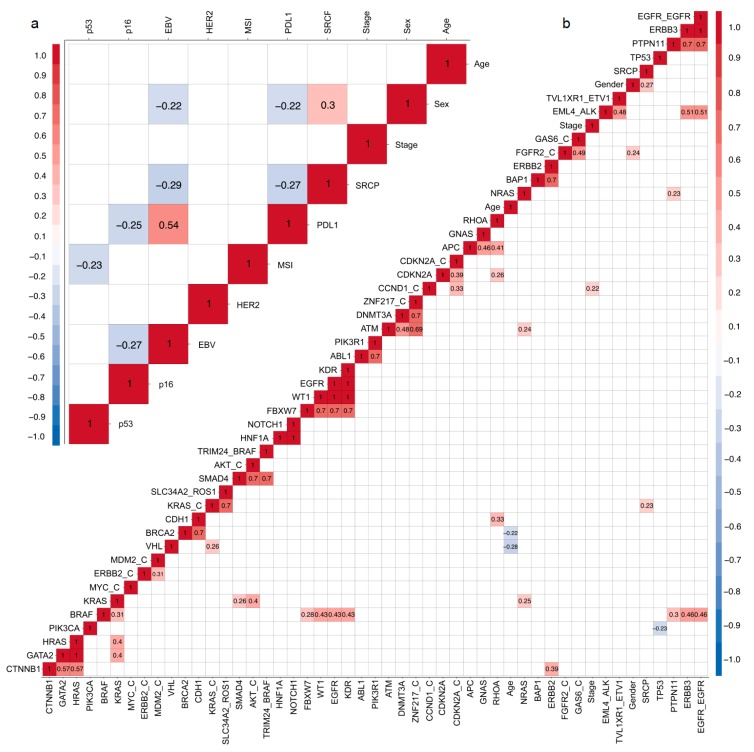
Kendall correlations between tissue microarray (TMA) and clinical data (**a**) or between NGS and clinical data (**b**). Significant correlations (*p* < 0.05) are indicated by colored squares; positive correlations are in red and negative in blue.

**Table 1 cancers-11-01275-t001:** Demographic and clinico-pathological characteristics of FORCE-1 study population (*n* = 224), tissue microarray (TMA)-analyzed subgroup (*n* = 90) and next generation sequencing (NGS) subgroup (*n* = 101).

Characteristic	FORCE-1 *n* (%)	TMA *n* (%)	NGS *n* (%)
**Gender**			
Male	142 (63.4)	58 (64.4)	67 (66.3)
Female	82 (36.6)	32 (35.6)	34 (33.7)
**Stage at diagnosis**			
I	30 (13.4)	8 (8.9)	9 (8.9)
II	57 (25.4)	24 (26.7)	31 (30.7)
III	88 (39.3)	46 (51.1)	49 (48.5)
IV	49 (21.9)	12 (13.3)	12 (11.9)
**ECOG Performance Status**			
0	69 (30.8)	25 (27.8)	29 (28.7)
1	69 (30.8)	27 (30.0)	26 (25.7)
2	6 (2.7)	2 (2.2)	3 (3.0)
≥3	1 (0.4)	1 (1.1)	1 (1.0)
NA	79 (35.3)	35 (38.8)	42 (41.6)
**Location of primary tumor**			
Distal esophagus and GEJ	49 (21.9)	18 (20.0)	24 (23.8)
Fundus	12 (5.3)	8 (8.9)	8 (7.9)
Corpus	86 (38.4)	28 (31.1)	32 (31.7)
Antrum	54 (24.1)	23 (25.6)	25 (24.8)
Pylorus	8 (3.6)	5 (5.6)	6 (5.9)
Multiple	9 (4.0)	4 (4.4)	3 (3.0)
NA	6 (2.7)	4 (4.4)	3 (3.0)
**Lauren histological type**			
Intestinal	76 (33.9)	27 (30.0)	34 (33.7)
Diffuse	61 (27.2)	32 (35.6)	30 (29.7)
Mixed	17 (7.6)	13 (14.4)	14 (13.9)
NA	70 (31.3)	18 (20.0)	23 (22.8)
**WHO histological type**			
Adenocarcinoma	171 (76.3)	71 (78.9)	80 (79.2)
Undifferentiated carcinoma	9 (4.0)	6 (6.7)	4 (4.0)
Adenosquamous cell carcinoma	3 (1.3)	3 (3.3)	3 (3.0)
NA	41 (18.3)	10 (11.1)	14 (13.9)
**Signet-ring cell presence**			
No	122 (54.5)	32 (35.6)	59 (58.4)
Yes	74 (33.0)	53 (58.9)	34 (33.7)
NA	28 (12.5)	5 (5.6)	8 (7.9)
**Comorbidities at diagnosis**			
Two or less	192 (85.7)	79 (87.8)	84 (83.2)
Three or more	32 (14.3)	11 (12.2)	17 (16.7)
**Age**			
Mean, median (range)	61.4, 62 (26–89)	62.7, 63 (26–89)	60.9, 62 (27–88)

*GEJ* gastroesophageal junction; *NA* not available.

**Table 2 cancers-11-01275-t002:** Immunohistochemistry tumor analysis (*n* = 90).

IHC Analysis	*n* (%)
**PDL-1 ^a^**	
Negative	64 (71.1)
Positive	26 (28.9)
**MSI+ (MMR deficient)**	13 (14.4)
MLH-1	
Intact	78 (86.7)
Lost	12 (13.3)
PMS-2	
Intact	78 (86.7)
Lost	12 (13.3)
MSH-2	
Intact	90 (100)
Lost	0
MSH-6	
Intact	88 (97.8)
Lost	2 (2.2)
**HER-2**	
Negative	78 (86.7)
Positive	12 (13.3)
**CISH-EBV**	
Negative	78 (86.7)
Positive	12 (13.3)
**p16**	
Absence	33 (36.7)
Presence	57 (63.3)
**p53**	
Intact	42 (46.7)
Mutated	48 (53.3)

**^a^** PDL-1 expression ≥ 10 by combined positive score (CPS). *IHC* immunohistochemistry, *MSI* microsatellite instability, *MLH-1* MutL protein homolog 1, *PMS-2* postmeiotic segregation increased 2; *MSH-2* MutS protein homolog 2; *MSH-6* MutS protein homolog 6; *HER2* human epidermal growth factor receptor 2; *CISH* chromogenic in situ hybridization; *EBV* Epstein-Barr virus; *NA* not available.

**Table 3 cancers-11-01275-t003:** Frequent gene alterations in FORCE-1patients.

Mutation Gene	Frequency *n* (%)	Mutation Gene	Frequency *n* (%)
SNVs		CNVs	
*TP53*	49 (48.51)	*MYC ^a^*	5 (4.95)
*PIK3CA*	15 (14.85)	*CCND1 ^a^*	4 (3.96)
*VHL*	6 (5.94)	*CCNE ^a^*	4 (3.96)
*NRAS*	7 (6.93)	*FGFR2 ^a^*	4 (3.96)
*KRAS*	6 (5.94)	*ERBB2 ^a^*	3 (2.97)
*BRAF*	5 (4.95)	*MDM2 ^a^*	3 (2.97)
*APC*	5 (4.95)	*CDKN2A ^b^*	2 (1.98)
*PTEN*	5 (4.95)	*KRAS ^a^*	2 (1.98)
*RHOA*	4 (3.96)	*AKT1 ^a^*	1 (0.99)
*CDKN2A*	3 (2.97)	*CDK6 ^a^*	1 (0.99)
*CTNNB1*	3 (2.97)	*GAS6 ^a^*	1 (0.99)
*ATM*	2 (1.98)	*ZNF217 ^a^*	1 (0.99)
*PIK3R1*	2 (1.98)		
*PTPN11*	2 (1.98)	**Fusions**	
*ERBB3*	1 (0.99)	*EML4_ALK*	4 (4.65)
*FBXW7*	2 (1.98)	*EGFR_EGFR*	1 (1.16)
*DNMT3A*	2 (1.98)	*SLC34A2_ROS1*	1 (1.16)
*SMAD4*	2 (1.98)	*TBL1XR1_ETV1*	1 (1.16)
*CDH1*	2 (1.98)	*TRIM24_BRAF*	1 (1.16)
*ERBB2*	2 (1.98)		

^a^ Amplification, ^b^ Deletion. Percentage (%) was calculated as the frequency of samples with said gene alteration (*n*) divided by the total of samples that passed DNA quality control (*n* = 101) for SNVs and CNVs, and RNA quality control (*n* = 86) for Fusions. *SNVs* Single nucleotide variations, *CNVs* Copy number variations.

**Table 4 cancers-11-01275-t004:** Most frequent mutations (>2.0% prevalence) found in FORCE-1 samples and comparison with different TCGA cohorts.

Gene	Aminoacidic Mutational Change	Total Samples *n*	101	100	30	295	66
		Function	Chilean FORCE-1 *n* (%)	UHK TCGA *n* (%)	UTOKIO TCGA *n* (%)	TCGA Nature 2014 *n* (%)	Brazil TCGA *n* (%)
*TP53*	R273C	Missense	4 (4.0%)	4 (4.0%)	NR	6 (2.0%)	1 (1.5%)
	R213 *	Nonsense	3 (3.0%)	1 (1.0%)	NR	5 (1.7%)	NR
	R175H	Missense	2 (2.0%)	NR	1 (3.3%)	6 (2.0%)	1 (1.5%)
	R248Q	Missense	2 (2.0%)	4 (4.0%)	NR	5 (1.7%)	3 (4.5%)
	R248W	Missense	2 (2.0%)	1 (1.0%)	NR	1 (0.3%)	NR
	P98S	Missense	2 (2.0%)	NR	NR	NR	NR
	Y220H	Missense	2 (2.0%)	NR	NR	NR	2 (3%)
	C242F	Missense	2 (2.0%)	NR	NR	NR	NR
*PIK3CA*	E542K	Missense	4 (4.0%)	NR	NR	5 (1.7%)	2 (3%)
	C378R	Missense	2 (2.0%)	NR	NR	1 (0.3%)	NR
	E545K	Missense	2 (2.0%)	NR	NR	11 (3.7%)	2 (3%)
	R88Q	Missense	2 (2.0%)	NR	NR	4 (1.4%)	1 (1.5%)
	T1025A	Missense	2 (2.0%)	NR	NR	NR	NR
*VHL*	S68L	Missense	6 (5.9%)	NR	NR	NR	NR
*NRAS*	G13V	Missense	5 (5.0%)	NR	NR	NR	NR
	G12D	Missense	2 (2.0%)	NR	NR	NR	NR
*KRAS*	G12D	Missense	3 (3.0%)	2 (2%)	NR	7 (2.4%)	1 (1.5%)
*BRAF*	D594G	Missense	3 (3.0%)	NR	NR	NR	NR
*RHOA*	Y42C	Missense	3 (3.0%)	NR	4 (13.3%)	3 (1%)	NR
*APC*	D156fs	Frameshift deletion	2 (2.0%)	NR	NR	NR	NR

*NR* Not reported. * indicates nonsense mutation.

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
