# Peer review of "High Proportion of Potential Candidates for Immunotherapy in a Chilean Cohort of Gastric Cancer Patients: Results of the FORCE1 Study"

_cancers, 2019, doi:10.3390/cancers11091275_

Round 1
Reviewer 1 Report
I recently received for review the article titled "High proportion of potential candidates for immunotherapy in a Chilean cohort of gastric cancer patients: results of the FORCE1 study" by Cordova-Delgado M et al.
The methods of the work have been explained in detail and the results that you presented support the conclusions of your work. I think that the title of the paper that you submitted somehow reduces the sheer number of results that you presented (the analysis is not only focused to search for candidates for immunotherapy but also for other potentially actionable targets in patients affected by gastric cancer).
It is interesting to notice that the results that you presented in a population of patients from South America with high prevalence of HP infection and gastric cancer, where 13% of gastric cancer samples that have been analysed have D-MMR status due to lack of IHC expression of hMLH1, is pretty much similar to previously published results conducted in other countries where increased gastric cancer risk is mainly related to HP infection (p.e. in Italy, as reported in Gastric Cancer. 2017 Jan;20(1):156-163, where the authors have found 14% D-MMR proportion in the subset of patients that have been analysed).
Another interesting point (that should be further discussed in the DISCUSSION section of the paper) is that, albeit in the D-MMR subgroup a greater proportion of patients with "intestinal" histology by Lauren were found, a smaller proportion was also diagnosed having "diffuse" histology. This seems to somehow be different from what is derived from TCGA analyses that usually suggest that "diffuse" histology gastric cancer samples should belong to the GS (genomically stable) subgroup more than to the MSI sugbroup. I would like to see some comment by the authors about this point.
In addition to that I would like to thank you for having made publicly available the results of the analyses in a way as to be able to reproduce them. Reading through the supplementary files I would like to point out at a few minor typos (such as in supplementary table 1 where it is lacking in the heading the + sign in the caption for MSI and where I would have liked that the stratification of patients in each subgroup by clinical characteristics should have been made by keeping in mind the number of patients by each subgroup instead of the overall number of patients in the whole cohort).
I think that these minor revision would add up to the scientific value of this work and that they would increase the interest in this topic.
Author Response
I recently received for review the article titled "High proportion of potential candidates for immunotherapy in a Chilean cohort of gastric cancer patients: results of the FORCE1 study" by Cordova-Delgado M et al.
1.The methods of the work have been explained in detail and the results that you presented support the conclusions of your work. I think that the title of the paper that you submitted somehow reduces the sheer number of results that you presented (the analysis is not only focused to search for candidates for immunotherapy but also for other potentially actionable targets in patients affected by gastric cancer)
R1. Thank you for reviewing our work and for your positive comments. We agree that the title is somehow restricted to immunotherapy candidates, however all authors also agree that the current title is more attractive to readers and highlights the most relevant finding of our study. We hope this is acceptable for the reviewer
2.It is interesting to notice that the results that you presented in a population of patients from South America with high prevalence of HP infection and gastric cancer, where 13% of gastric cancer samples that have been analysed have D-MMR status due to lack of IHC expression of hMLH1, is pretty much similar to previously published results conducted in other countries where increased gastric cancer risk is mainly related to HP infection (p.e. in Italy, as reported in Gastric Cancer. 2017 Jan;20(1):156-163, where the authors have found 14% D-MMR proportion in the subset of patients that have been analysed).
R2. Interesting indeed, we have incorporated the suggested reference into the discussion section of the revised manuscript in order to make the point that our 13% of D-MMR is similar to previous studies (14%).
3.Another interesting point (that should be further discussed in the DISCUSSION section of the paper) is that, albeit in the D-MMR subgroup a greater proportion of patients with "intestinal" histology by Lauren were found, a smaller proportion was also diagnosed having "diffuse" histology. This seems to somehow be different from what is derived from TCGA analyses that usually suggest that "diffuse" histology gastric cancer samples should belong to the GS (genomically stable) subgroup more than to the MSI sugbroup. I would like to see some comment by the authors about this point.
R3. Very interesting point. There are a number of differences between our study and the TCGA that could explain the percentage of Diffuse cases in our MSI-H subgroup. First of all, TCGA used Whole-Exome Sequencing and actual MSI to define this subgroup, in contrast we used d-MMR to identify potential MSI-H which might be an overestimation. Secondly, derived from this our study could not define GS or CIN subgroups of patients, therefore we cannot estimate the % of diffuse among GS. Still, TCGA indicates that there is a 73% of diffuse among GS and a about 10% among MSI-H. Evidently, histological classifications such as Lauren are inaccurate and more comprehensive systems such as a molecular classifications may provide better prognostic information and hopefully actionable targets. As the reviewer requested some of these ideas are briefly discussed in the discussion section of the revised manuscript
4.In addition to that I would like to thank you for having made publicly available the results of the analyses in a way as to be able to reproduce them. Reading through the supplementary files I would like to point out at a few minor typos (such as in supplementary table 1 where it is lacking in the heading the + sign in the caption for MSI and where I would have liked that the stratification of patients in each subgroup by clinical characteristics should have been made by keeping in mind the number of patients by each subgroup instead of the overall number of patients in the whole cohort).
R4. Thank you for noticing this. We checked supplementary material for typos, added a missing + sign in MSI in Table S1. Regarding subgroups we recalculated numbers and added this instead the overall of patients
5. I think that these minor revision would add up to the scientific value of this work and that they would increase the interest in this topic.
R5. Thank you again for your comments and valuable suggestions. We hope this new revised version is suitable for publication
Reviewer 2 Report
This is a very interesting manuscript aiming to evaluate the molecular profile of gastric cancer among the Chilean population. This is the first paper to study genetic variants in South American patients who are at high risk of developing and dying from the disease. The TMA cohort and NGS cohort are enriched for stage III patients, how does this bias you’re results?
I would like to suggest minor revisions and comments to the authors:
Results:
Consider adding the average age of your cohort population to table 1. Define TMA (line 157). It would be helpful to add that you evaluated markers by immunohistochemistry (line 159). Line 160: extra “A” in text. I particularly liked how the authors highlighted the actionable mutations
Discussion:
Define TMB (line 311). Does high-TMB relate to patients that are MSI-High? Does the high rate of H pylori influence any of these results?
Methods:
Describe pathology examination of TMA and EBV status. Which marker was used to determine EBV status (EBERs)? (Line 392) Describe how HER2 status was determined? Describe how PD-L1 positive patients were determined? More that 5% of the tumor positive, include immune cells? More description in the methods would be helpful.
Conclusions:
You mention a high proportion of patients could benefit from use of immunotherapeutic drugs, it could be helpful to remind your reader which patients the authors believe could benefit: EBV+ patients (13%), MSI high (13%), high-TMB (20%).
Author Response
1-This is a very interesting manuscript aiming to evaluate the molecular profile of gastric cancer among the Chilean population. This is the first paper to study genetic variants in South American patients who are at high risk of developing and dying from the disease. The TMA cohort and NGS cohort are enriched for stage III patients, how does this bias you’re results?
R1. Thank you for taking the time to review our work and for your kind comments. Regarding the proportion of stage III you are correct, some of our team members are gastric surgeons and given that these patients go through surgery the proportion of stage III may be overrepresented, this could be a registration bias of our study. We have incorporated this comment as a limitation of our study into the discussion section of the revised manuscript.
I would like to suggest minor revisions and comments to the authors:
Results:
2. Consider adding the average age of your cohort population to table 1. Define TMA (line 157). It would be helpful to add that you evaluated markers by immunohistochemistry (line 159). Line 160: extra “A” in text. I particularly liked how the authors highlighted the actionable mutations
R2. Done. We have added average age into Table 1 as requested. Also we defined TMA and specified we measured markers by IHC. We eliminated the extra A. Thank you for your comment
Discussion:
3. Define TMB (line 311). Does high-TMB relate to patients that are MSI-High? Does the high rate of H pylori influence any of these results?
R3. We defined TMB in the revised manuscript. Good question, we found that 40% (4/10) of MSI patients were also high-TMB. In contrast, only 11.9% (7/59) of MSS were high-TMB. A new panel in supplementary fig.2 has been added to illustrate this point in the new revised version of the manuscript. Regarding HP infection, unfortunately our study did not determine HP rates.
Methods:
4. Describe pathology examination of TMA and EBV status. Which marker was used to determine EBV status (EBERs)? (Line 392) Describe how HER2 status was determined? Describe how PD-L1 positive patients were determined? More that 5% of the tumor positive, include immune cells? More description in the methods would be helpful.
R4. As requested, this information has been added and we have expanded the methods section in the revised manuscript.
Conclusions:
5. You mention a high proportion of patients could benefit from use of immunotherapeutic drugs, it could be helpful to remind your reader which patients the authors believe could benefit: EBV+ patients (13%), MSI high (13%), high-TMB (20%).
R5. Following this recommendation we have added the patient subsets (and their proportions) that could benefit from immunotherapy in the CONCLUSION section